# Impact of two endotracheal tube fixation on the incidence of peri-oral lesions: Elastic adhesive strips versus cord in a protective sheath. Study protocol for a cluster cross-over randomized trial

**Vanessa Zinzoni**[1]\*, **Lucie Planche**[2], **Sophie Le Potier**[3], **Laurence Robin**[4], **Cécile Le Parco**[5], **Philippe Terrat**[6], **Marie-Hélène Leroyer**[7], **Romain Atger**[8], **Jérôme E. Dauvergne**[9,10], **Lucie Muller**[11], **Laetitia Fontaine**[12], **Célina Morand**[13], **Pascaline Dennemont**[14], **Ophélie Paillard**[15], **Servane Vastral**[16], **Baptiste Dardaine**[17], **Sylvie Le Guillou**[18], **Natacha Maquigneau**[1], **Stéphanie Martin**[2], **Jean-Claude Lachérade**[1]

**1** Service de Médecine Intensive Réanimation, Centre Hospitalier Départemental de Vendée, La Roche-sur-Yon, France, **2** Unité de Recherche Clinique, Centre Hospitalier Départemental de Vendée, La Roche-sur-Yon, France, **3** Service de Réanimation Polyvalente, Centre Hospitalier Bretagne Sud, Lorient, France, **4** Service de Réanimation Polyvalente, Centre Hospitalier d'Angoulême, Angoulême, France, **5** Service de Réanimation Polyvalente, Centre Hospitalier Victor Dupouy, Argenteuil, France, **6** Service de Réanimation, Centre Hospitalier de La Rochelle, La Rochelle, France, **7** Service de Réanimation Médico-Chirurgicale, Centre Hospitalier du Mans, Le Mans, France, **8** Service de Médecine Intensive Réanimation, Centre Hospitalier Universitaire de Nantes, Nantes, France, **9** Service d'anesthésie-réanimation, INSERM, CIC 1413, Hôpital Laënnec, Nantes Université, CHU Nantes, Nantes, France, **10** Institut du Thorax, CNRS, INSERM, Nantes Université, CHU Nantes, Nantes, France, **11** Service de Médecine Intensive Réanimation, Centre Hospitalier Régional d'Orléans, Orléans, France, **12** Service de Réanimation Médico-Chirurgicale, Centre Hospitalier Intercommunal de Poissy/Saint Germain-en-Laye, Poissy, France, **13** Service de Médecine Intensive Réanimation, Centre Hospitalier Universitaire de Poitiers, Poitiers, France, **14** Service de Réanimation Polyvalente, Centre Hospitalier Universitaire de la Réunion, Saint-Pierre, La Réunion, France, **15** Service de Neuro-réanimation, Centre Hospitalier Universitaire de la Réunion, Saint-Pierre, La Réunion, France, **16** Service de Médecine Intensive Réanimation, Centre Hospitalier de Saint-Nazaire, Saint-Nazaire, France, **17** Service de Médecine Intensive Réanimation, Centre Hospitalier Régional Universitaire de Tours, Tours, France, **18** Service Anesthésie-Réanimation Chirurgicale, Centre Hospitalier Universitaire de Nantes, Nantes, France

\* vanessa.zinzoni@ght85.fr

## Abstract

### Background

Endotracheal tube fixation in ventilated patients must be appropriate to ensure security during mechanical ventilation and prevent skin lesions. The incidence of endotracheal tube-caused pressure ulcers ranges from 7% to 45%. Various endotracheal tube fixations are used in intensive care units (ICUs) worldwide. By pressure exercised on the skin, these systems could lead to mucosal and skin peri-oral lesions. The main objective of this study is to evaluate the impact of the two fixation systems most commonly used in French ICUs (adhesive elastic band versus fixation cord with PolyVinyl Chloride (PVC) sheath) on the incidence of these peri-oral skin lesions.

**Data Availability Statement:** This study protocol article does not report data and the data availability policy is consequently not applicable.

**Funding:** This study was financially supported by the French Ministry of Health (Programme Hospitalier de Recherche Infirmière et Paramédicale) [https://sante.gouv.fr.] in the form of a grant (2019, PHRIP-19-0140) received by VZ. No additional external funding was received for this study. The funder had no role in study design, data collection and analysis, decision to publish, or preparation of the manuscript.

**Competing interests:** The authors have declared that no competing interests exist.

## Methods

This studyis a multicenter, open-label, controlled, superiority, cluster cross-over randomized trial. 768 patients will be recruited in the 16 ICUs involved. The inclusion of patients will be carried out over two 12-month periods. Each site begins with one of the evaluated fixation systems: elastic adhesive tape or cord associated with a protective sheath. After a 4-month break, each site switches to the other fixation system. The primary outcome is the development of at least one peri-oral lesion during the first ten days of maintaining an orally inserted endotracheal tube. The presence of lesions is assessed by a blinded adjudication committee using photographs taken daily.

## Discussion

This study is the first multicenter, randomized trial designed to evaluate the impact of elastic adhesive tape versus fixation cord with PVC sheath on the incidence of peri-oral lesions. The results will provide data which could change and standardize care practices.

## Trial registration

https://www.clinicaltrials.gov. Reference number: NCT04819425.

## Background

### Rationale

Nearly 100,000 French patients per year hospitalized in an intensive care unit (ICU) for more than 48 hours (extrapolation of data from the REA-RAISIN, Nosocomial Infections Alert, Investigation and surveillance network of French ICU) require invasive respiratory assistance with the placement of an endotracheal tube (ETT), most often introduced via the oro-tracheal route. Patients hospitalized in ICU are at risk of developing pressure ulcers, and prevention is an aspect of daily nursing care [1]. With the development of new medical devices (MD), the prevention of medical device-related pressure injuries (MDPI) has become a new focus for improving the management of intensive care patients. An MDPI is defined as an area of localized injury to the skin or underlying tissue as a result of sustained pressure from devices designed and implemented for diagnostic or therapeutic purposes. The resulting tissue lesion has the shape of the medical device and has the propensity to progress rapidly due to minimal fatty tissue at the various sites of ulceration [2, 3]. The most common MDPIs identified by ICU nurses are those related to endotracheal tubes [2, 4, 5].

The ETTs are attached to the face to ensure that it is maintained in an optimal position for effective ventilation. The correct positioning of the endotracheal tube is first checked by chest X-ray and then verified daily by the nurse using graduated markers. The fixation systems are numerous and differ according to the ICU considered. However, this fixation of the ETT to the face has the counterpart of generating excessive pressure on the underlying skin, source of mucocutaneous lesions of the peri-oral region [6]. Beyond the pain experienced, the resulting lesions can be unsightly, sometimes resulting in scarring. This can have a psychological impact on the patients, with an alteration of the self-image causing anxiety that can go as far as social isolation of the patients. This can also give the family a negative image of the care given to the patient and goes against the notion of "taking care" which is at the heart of the nurse's role.

Several types of lesions are encountered in the peri-oral area: skin lesions, mucosal lesions, and shearing/cuts. The pressure associated with the presence of ETT fixation systems is exerted over a prolonged period (on average 7 to 10 days for patients ventilated more than 48 hours). In addition, patients are frequently unable to move the tube in case of discomfort or to express their pain or discomfort (initial coma, deep sedation, etc.). There are risk factors for the development of pressure ulcers in critically ill patients such as sedation, decreased tissue perfusion, malnutrition and patients on vasopressors [1]. The reported ICU ETT-caused pressure ulcer incidence rates range from 7% to as high as 45% [7]. A meta-analysis states that the incidence of pressure injury was 4.2% for orotracheal tubes [8]. Pressure ulcers become clinically visible between 2 and 13 days after intubation [7]. Nurses currently use the National Pressure Ulcer Advisory Panel (NPUAP) scale, a pressure ulcer scale to assess and monitor the evolution of lesions. A specific scale, the "ROMPIS" scale has been developed to evaluate mucosal lesions which differ in nature, histological characteristics and healing process [6, 9].

Injuries related to these fixation systems develop for several reasons. First, the endotracheal tube is usually made of a rigid material that can cause friction or pressure on the underlying soft tissue. The adhesive tapes used to secure these devices can also irritate sensitive skin, and the cord can cause shearing or cutting, especially in the presence of edema, which is often the case in the ICU. Several fixation systems are available: simple fixation methods not specifically developed as endotracheal tube fixation systems (plasters, adhesive elastic bands, etc.), or developed exclusively for this use (cord, cord plus sheath, foam), or sophisticated systems developed by the pharmaceutical industry (such as AnchorFast, Marpac 320, Stabilock. . . ..). Their purpose is to minimize the movement of the tube, to be able to be installed quickly and to create as little skin and mucous membrane damage as possible for the patient [10]. The impact of sophisticated devices developed by the pharmaceutical industry on the development of peri-oral lesions varies according to the studies. They are sometimes associated with an increase in peri-oral lesions (before-and-after study, published in 2018, on a group of 1100 patients evaluating a simple cord fixation versus the " AnchorFast ™") [11] or sometimes associated with a decrease in peri-oral lesions (single- center prospective study published in May 2019 comparing the "AnchorFast ™" system and adhesive strips) [12], (comparative non-randomized study, published in 2020, on a group of 155 patients comparing cloth tape with Anchorfast Hollister) [13]. A randomized monocenter trial was conducted on 60 Turkish patients in 2020, comparing bandage fixation and ETT tube holder used as ETT securement. Bandage fixation was better than the endotracheal tube holder technique in terms of both the pressure sore risk score difference and the tendency to fall, dislocate or remain stable, according to the Braden Scale for Predicting Pressure Sore Risk assessment results of the first and fourth days [14]. Nevertheless, fixation systems developed by the pharmaceutical industry exert more pressure on the facial skin as suggested by a mannequin study comparing 10 sophisticated industrial systems and 6 "simple" systems [10]. Regarding preventive measures for the prevention of unplanned endotracheal extubation in the ICU, a recent review concluded that an optimal method of endotracheal tube fixation has not been established [15].

The French Society for Intensive Care Medicine (Société de Réanimation de Langue Française (SRLF) carried out a prospective observational study in 2011 (IDEFIX Study, data not published, but provided during the French Intensive Care congress in 2012, see S1 Appendix) which also showed the heterogeneity of practices with the use of different fixation systems. The study indicated the predominant use of the fixation cord (more or less associated with a protective sheath) before that of an elastic adhesive band. In early 2019, we carried out a survey in which 29 French ICUs declared their practice. Thus, among the 6 different fixation systems used, 2 stand out: the elastic adhesive tape (34.5%) and the cord associated with a protective sheath (24%). Simple fixation systems are therefore more often used in French ICUs than

sophisticated systems developed by the pharmaceutical industry. The divergent results concerning the cutaneous-mucosal impact of sophisticated fixation systems partly explain their low use in French intensive care units. Their high cost compared to simple systems is certainly also an obstacle to their use. There is currently no formal recommendation for the use of a particular endotracheal tube fixation system. This encourages the heterogeneity of fixation practices in intensive care units.

## Objectives and study hypotheses

The main objective of this study is to evaluate the impact of the two fixation systems most commonly used in intensive care units in France (adhesive elastic band versus fixation cord with PolyVinyl Chloride (PVC) sheath) on the incidence of peri-oral skin lesions. The study hypothesis is that elastic adhesive tape fixation decreases the risk of developing a peri-oral lesion before the 10th day of maintaining the endotracheal tube compared with ETT fixation with a cord with a PVC sheath.

However, the authors bear also in mind a larger objective which is to find a reliable endotracheal tube fixation leading to less peri-oral tube but which should also allow the ETT to be maintained in an optimal position.Consequently, one of the secondary objective is to check the efficacy of the device in maintaining the endotracheal tube in terms of self-extubation and repositioning of the tube.

The other secondary objectives are to compare the time to the first peri-oral lesion, to compare the intensity of mucosal and skin peri-oral lesions assessed by nurses, to assess the incidence of peri-oral lesions and to evaluate the impact of the intervention on the maximum number of lesions per day.

## Materials and methods

### Design

This study is a multicenter, open-label, controlled, superiority, cluster cross-over randomized trial. The study is performed in 16 French medical, neuro or medico-surgical ICUs. Seven of them are located in university hospitals and nine are in general hospitals. The inclusion of patients will be carried out over two 12-month periods. Each site begins with one of the evaluated fixation systems: elastic adhesive tape or cord associated with a protective sheath. After a 4-month break, each site switches to the other fixation system.

### Recruitment and eligibility

Patients are recruited in French ICUs participating in the study.

**Inclusion criteria.** Adult patient ($\geq$18 years), requiring mechanical ventilation for an estimated duration > 48 hours and treated with vasopressors, are eligible for this study.

Patients or family members should have given consent to participate in the study or should be included with the emergency procedure in the absence of contactable family members.

**Exclusion criteria.** Patients are not included if they meet one or more of the following criteria:

- Pre-existing facial lesions on arrival in the ICU on the path of the endotracheal tube fixation

- Admitted intubated following transfer from another ICU

- Nasotracheal intubation

- Patient in isolation for suspected COVID or clinically proven COVID

- Patient admitted with tracheostomy

- Moribund (High probability of death within 48 hours of inclusion)

- Pregnant, nursing, and parturient women

- Lack of social security affiliation

- Incapacitated adult (under guardianship, curatorship) or deprived of liberty by court order

## Randomization

The chronology of the two periods has been assigned for each center (cluster) according to a randomization list before the start of the study.

The study focuses on professional practices. Some centers did not use the studied fixation systems and have to be trained in their use, which justifies a collective randomization unit for the proper implementation of this new care practice and a reduction of the risk of error.

To ensure homogeneity of the practices of fixation of endotracheal tubes within a given ICU during each of the two inclusion periods, all ventilated patients, regardless of the their status, included or not in the study, have a tube fixated with the fixation system assigned to the center during this period.

## Intervention

This study is a cluster and cross-over randomization trial. Therefore, each center uses one or the other fixation system alternately, and according to the randomization list, for a period of 12 months.

Before the fixation systems were dispatched to the centres, all medical and paramedical staff at each investigation center are briefed on the fixation methods to be used during the period.

One month before the start of inclusion of each period, the investigating centers are equipped with the fixation devices allocated by the randomization list. In this way, the fixation procedure is implemented into the management of patients of these centers, and that the paramedical teams are able to familiarize themselves with the technique. The previous ETT fixation system used in the center should not be available anymore for ETT fixation of ventilated patients. The only ETT fixation system available in the center is the one assigned by randomization.

The schedule of enrolment, interventions and assessments is detailed in Fig 1 and a study plan in Fig 2.

**Description of fixing techniques.**   *Technique A*: *Fixation with elastic adhesive tape*. The available tape has a width of 3 cm.

1. Cut the elastic tape in 4 and then in 2 in the width to obtain 8 strips. The length obtained is thus 62–63 cm and about 1.5 cm wide.

2. Fix the adhesive strip on the patient's face (opposite side to the endotracheal tube) and then two turns around the endotracheal tube will be made. The rest of the tape should be attached to the other side of the face (endotracheal tube side).

3. Keep the plastic bandage on the adhesive bandage as long as it passes over the neck so that it does not stick to the hair.

4. Finally, replace the end of the tape on the part already attached to the patient.

| | STUDY PERIOD | | | | |
| --- | --- | --- | --- | --- | --- |
| | **Enrolment** | **Allocation** | **Post-allocation** | | |
| **TIMEPOINT** | D0 | **D0** | **D0** | **D10** | **D28** |
| **ENROLMENT:** | | | | | |
| **Eligibility screen** | X | | | | |
| **Informed consent** | X | | | | |
| **Allocation** | | X | | | |
| **INTERVENTIONS:** | | | | | |
| *Elastic adhesive strips* | | | ●————————● | | |
| *Cord in a protective sheath* | | | ●————————● | | |
| **ASSESSMENTS:** | | | | | |
| *Daily Assessment of peri-oral lesions* : *Paramedical examination* | | | ●————————● | | |
| *Photographs of lips for adjudication committee* | | | ●——————● | | |
| *Daily collection of ETT tube mobilizations* | | | ●————————● | | |
| *Daily collection of ETT tube repositioning* | | | ●————————● | | |
| *Daily collection of re-intubation* | | | ●————————● | | |
| *Daily collection of ETT tube repositioning* | | | ●————————● | | |
| *Collection of undesirable events* | | | ●————————● | | |

**Fig 1. Schedule of enrolment, interventions and assessments.** D : Day.

Technique B: Fixation by cord and PVC sheath.

1. Pull the loop (in the middle of the fixation) to loosen it

2. Place the loop around the tube from underneath.

3. Pass the PVC part of the fixation through the loop.

4. Pull the ends of the cord out of the sheath to tighten the loop around the probe.

5. Tie a knot with both ends of the cord so that the fixation is stable around the patient's head.

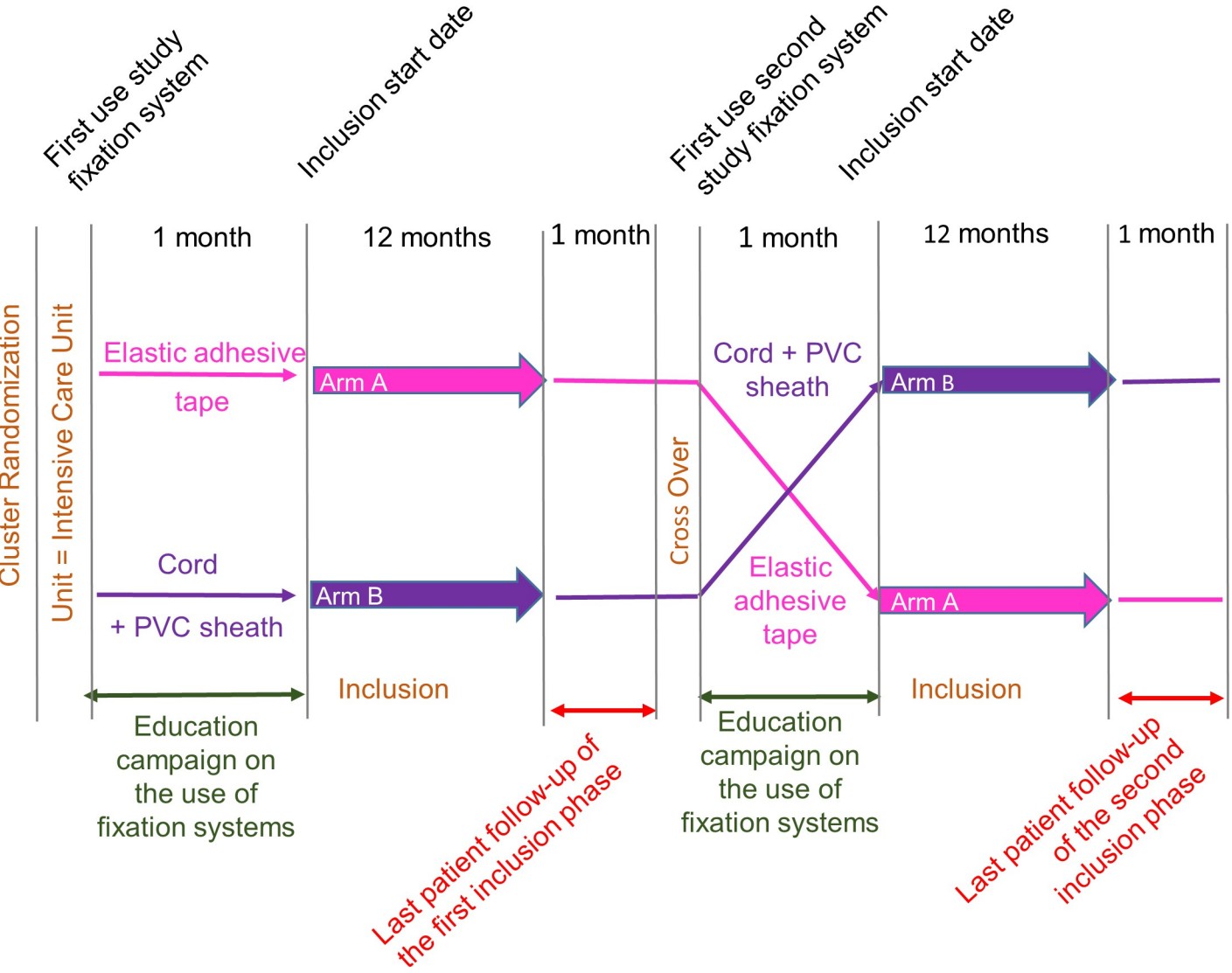

**Fig 2. Study plan.**

Photos in Fig 3 show both methods of ETT fixation system used in the study.

**First use of ETT fixation system.** After having ensured the consent of the patient or his/her relatives or having followed the emergency inclusion procedure, the investigator can then include the patient before the day 1 photograph. If the patient is already intubated before ICU admission, the ETT fixation has to be systematically changed to install the fixation system used in the ICU depending on the study period considered. Day 0 is then defined as the first day of installation of the ETT fixation system (cord + sheath or elastic adhesive strips) provided for the study.

**Follow up.** The nurse in charge of the patient has to change the fixation system at least once a day and if necessary during the day in case of soiling or examinations. Duoderm-type dressings used to prevent pressure sores under the ETT are not accepted since this is the main topic of the study. However, it can be applied as a curative measure if a lesion appears. A verification of the correct positioning of the ETT thanks to the graduated marker is systematically

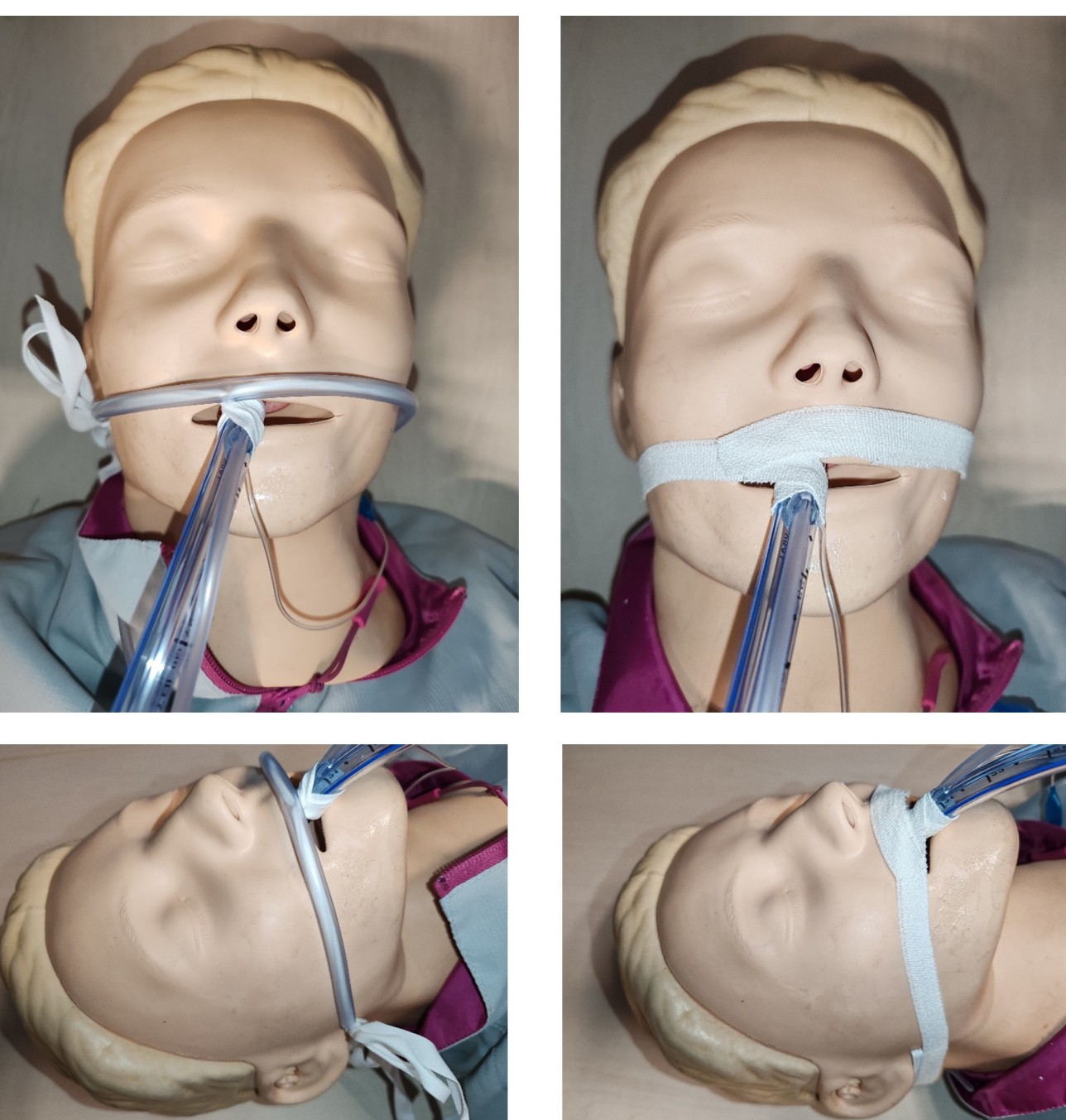

**Fig 3. Two methods of ETT fixation system used in the study.** On the left: fixation by cord and PVC sheath, on the right: fixation with elastic adhesive tape.

carried out once a day. If repositioning is necessary, the nurse will note the number of centimeters. The mobilization of the tube (change of side) is done according to the usual practices of the service.

During the two recruitment periods, a notebook is provided to the paramedical staff and includes the follow-up of each change of fixation system made during the stay in the ICU. Nurses in charge of the patient record in the notebook the presence of vasopressors, prone

positioning, or the presence of sedation, which are risk factors for skin pressure ulcer development in ICU patients. The number of mobilizations, interventions on the fixation system and the number of fixation system changes are also recorded daily. The presence or absence of lesions is assessed daily by the nurse in charge of the patient. She also assesses the type of lesion: cutaneous or mucous. Depending on the type, she will use the appropriate scale to assess the grade: the NPUAP scale (cutaneous) or/and the ROMPIS scale (mucous).The follow-up by the nurse's notebook will stop at the final extubation or the discharge from the ICU if the patient is discharged intubated from the ICU before D28 or at most D28.

Two pictures (one with the lower lip visible and one with the upper lip visible) are taken every day by the nurses from the day of the first fixation change = D0 until D10. In no case, the whole face of the patient is visible in the photo. The photos are taken without the fixation visible to allow the blind to be kept for the adjudication committee.

**Adjudication committee.**    The committee is composed of dermatologists and two nurses experienced in assessing pressure sores. They are not otherwise involved in the trial.

Each expert has to record the presence or not of a perioral lesion between D0 and D10.

For the primary endpoint, we selected a daily photo assessment until D10. The choice of this time frame was based on:

- the time of appearance of peri-oral lesions (more than 50% of them appear during the first 5 days) (IDEFIX trial, S1 Appendix),

- the average duration of ventilation for patients who require invasive respiratory assistance for more than 48 hours,

- risk factors for pressure sores which are essentially present at the beginning of the management of patients in shock (oedema, sedation, decrease in tissue perfusion, undernutrition, vasopressor treatments),

- the fact that the number of photography to be assessed by the adjudication committee should be limited to make the review process feasible.

**The 4-month break between period.**    An interruption period of 4 months between both periods is planned. This interim period allows for the completion of the 28-day follow-up of patients from the first period without changing the fixation system used in the center. For the next two months, centers remain with the fixation systems of the first period or could use their usual system if they wish. During the month preceding the beginning of the second inclusion period, centers change their ETT fixation system for the second fixation system of the project and compulsorily use it. The first system used in the center should not be available anymore for ETT fixation of ventilated patients.

## Outcomes measures and timelines

**Primary outcome measure.**    The primary endpoint is assessing the appearance of at least one perioral lesion during the first ten days of maintaining the orally inserted endotracheal tube. The perioral lesion is validated on photographs by an independent review committee.

**Secondary outcome measures.**    The secondary endpoints are:

- Number of self-extubations

- Number of significant repositioning of the tube performed following a movement of more than 2 cm from the initially set marker.

- Time to the first peri-oral lesion from the time the patient is intubated until extubation.

- Assessment of the grade of cutaneous peri-oral lesions, by the nurses, using the NPUAP scale. If several lesions co-exist, the lesion with the most severe grade will be retained.

- Evaluation of the intensity of the mucosal peri-oral lesions, by the nurses, using the ROMPIS scale. If several lesions co-exist, the lesion with the most severe grade will be retained.

- Number of patients with at least one peri-oral lesion per 1000 ventilation days

- Maximum number of lesions per day

## Data collection

**Adjudication committee.** To assess the primary criterion, the peri-oral lesions are validated on photographs by an independent review committee. Several dermatologists and 2 nurses, independent of the study, compose the adjudication committee. Each file (including all the photos of the same patient) are randomly transmitted to each member of the adjudication committee. Each member rules on the presence or absence of lesion and their type, cutaneous or mucosal.

Secondary criteria include the assessment of the grade of lesions using two scales : one for the cutaneous lesions (NPUAP) and one for the mucous (ROMPIS). This assessment is performed by the nurses at the bed of the patient and not by the adjudication committee. Indeed, the assessment of the grade is very difficult on photos as stated by Jesada et al. [16] "A digital photograph alone cannot reliably convey the characteristics of a pressure ulcer".

**National Pressure Ulcer Advisory Panel's Stages Scale (NPUAP Scale).** This scale assesses pressure sores according to 4 stages graduated from 1 to 4. Stage 1 is the least severe [17]. It is widely used in France.

**Reaper Oral Mucosa Pressure Injury Scale (ROMPIS Scale).** The second scale used is The Reaper Oral Mucosa Pressure Injury Scale (ROMPIS) for mucosal lesions. First published in 2017 [9], the ROMPIS scale is the only scale to assess mucosal injury in this setting. ROMPIS is graduated in 3 stages.

Other secondary criteria are evaluated using nurses' notebooks filled daily or medical record.

## Data management

An electronic Case Report Form (eCRF) is created for each patient summarizing all study data. The eCRF will include the data needed to confirm compliance with the protocol and all data needed for statistical analyses; it will allow the identification of any major deviations from the protocol. The pictures of the lips are also downloaded in the eCRF by the site's team. Adjudication committee members have access to these pictures online but not to any clinical data.

Subjects that are included in the study, are assigned a unique study number upon their registration in ENNOV Clinical (eCRF software). Participants are only recorded by their participant number. The subject identification list is safeguarded by the site. The name and any other directly identifying details are not included in the study database.

Data monitoring is carried out by employees of the department of the clinical research unit of Vendee Hospital to ensure that the eCRF are completed accurately, to review informed consents and the followed process, to check on recruitment status, protocol deviations/violations, good clinical practice (GCP) compatibility.

## Data analysis

**Sample size calculation.** The study will take place in 16 centers (clusters) over two 12-month periods.

According to the IDEFIX study (data not published, but provided during the French Intensive Care Congress in 2012, S1 Appendix), we hypothesize that 35% of patients with a "cord and sleeve" fixation system have a peri-oral lesion compared to 20% with the elastic adhesive tape system.

The calculation of the number of subjects needed must take into account the design of the study and thus the two intra-class and inter-class correlation coefficients [18].

As we have no data on the estimation of correlation coefficients, we have retained, for the calculation of the number of subjects, a "moderate" intra-class correlation coefficient and therefore defined at 0.1, and as recommended, a within-cluster between-period correlation coefficient equal to half of the intra-class correlation coefficient, i.e. 0.05 [19].

With 90% power and 5% alpha level, we calculated that each of the 16 centers would need to include an average of 24 patients per period.

A total of 768 patients are to be included.

**Statistical analysis planned.** The statistical analysis report will be written according to the recommendations proposed by Arnup et al. [20] on cross-over cluster trials.

To account for the study design, all criteria (primary and secondary) will be compared using hierarchical generalized linear models. The type of fixation, the period and the center (= cluster) will be taken into account in the model as fixed effects; the center period interaction will be taken into account as a random effect [21].

*Main criterion.* The percentage of patients with at least one peri-oral lesion will be estimated with a 95% confidence interval for the 2 types of fixation. A logistic regression model taking into account the duration of the patient's endotracheal tube retention and the study design will be applied. The within-cluster within-period coefficient correlation and the within-cluster between-period correlation will be reported.

*Secondary criteria.* The number of self-extubations and repositioning of the tube will be estimated and compared using a Poisson regression model taking into account the study design. An offset of the logarithm of the intubation time will be included in the model.

The time to the first peri-oral lesion will be estimated using a random-effects survival model to account for the study design.

The grade of skin lesions (NPUAP scale) and mucosal lesions (Rompis scale) will be collected for each lesion.

An ordinal logistic regression model controlling for intubation time and study design will be applied to compare the 2 types of fixation.

The incidence of peri-oral lesions (expressed per 1000 ventilator days) will be calculated and described in each group.

The maximum number of injuries per day will be estimated and compared using a linear model taking into account the study design.

## Ethics

The study protocol and patient information form were approved by the ethics committee of Nantes (Comité de Protection des Personnes Ouest IV) on 18 May 2020. Reference number : 20.02.21.63332.

According to French law and with approval from the ethics committee, this type of study does not need a written consent but an oral consent. Information and oral consent must be sought before any inclusion in the study. It should be carried out with the patients or, if

impossible, with their relatives. If the trusted person or the relative is physically absent, an emergency procedure is available to include the patient. However, in this particular case, both the relative and the patient should be informed as soon as possible. The information note should be given to them and their oral consent should be collected by the investigator and documented in patient's medical file. This procedure was approved by the ethics committee of Nantes. The first patient was included on 28 June 2021.

## Discussion

To date, there is currently no formal recommendation for the use of a particular endotracheal tube fixation system. This trial is the first randomized multicenter study comparing two simple fixation systems commonly used in ICU.

Our main criterion is the appearance of at least one perioral lesion during the first ten days of maintaining the orally inserted endotracheal tube. In order to homogenize the population and create the most favorable conditions for comparing the appearance of lesions, the study focuses on patients receiving vasopressors. Indeed vasopressor support therapy is a risk factor for pressure ulcer development [22]. It can increase the risk by a factor of 6 [23].

The design of the study is a cluster cross-over randomized design. The studied intervention goes beyond the simple use of a specific fixation system: it affects practices. Its design in cluster should allow a good implementation of the use of each fixation system. It is worth having a unique ETT fixation system during the period. Each ICU team will be trained and will use only one fixation system per period. It should reduce the risk of error or poor implementation of the system due to the coexistence of several systems in the same department. The cross-over design of the study is justified by the gain of power compared to a parallel group trial design and by the gain in comparability of the groups that could be expected compared to a simple cluster trial. It is also justified by the absence of possible residual effect of the study intervention. Indeed, the ETT fixation systems will be different in each of both periods. At the end of the period, the previous fixation systems system will be removed from the arsenal of equipment available in the department.

The main criterion chosen focused on perioral lesions. Due to the fact that the intervention could not be blinded, this main criterion is assessed in blind by an independent committee.

This criterion was chosen rather than the number of self-extubation or mobilization for the following reasons:

- The study is conducted by nurses and the prevention of the onset of pressure sores on the face is part of daily nursing care and is a real concern for them.

- Self-extubation can be linked to the fixation system but is mainly patient-dependent. Many patients are agitated when they begin to wake and this can lead to self-extubation whatever the attachment system.

However, securing the ETT with a suitable fixation system to prevent unplanned extubation is a matter that needs to be addressed. In their recent review, Wu and al [15] recommend carrying out more randomized controlled trials related to ETT fixation. The present study could provide data to make progress on that topic

An independent safety committee (ISC) board has been constituted after the start of the study.

There was no requirement for this committee to be constituted at the beginning of the study, given the nature of the study according to French Law. Sponsor transmitted a "New Safety Fact declaration" to the French authority, ANSM (Agence de sécurité du medicament et des produits de santé) on 22/11/2021. Indeed, some centers reported self-extubation attributed

to the fixation system used in the study. Consequently, sponsor felts it was important to analyze the frequency of events classified as "undesirable" that occurred between the 2 arms of the study, in particular self-extubations and repositioning of the endotracheal tube and constitute an ISC. The ISC is composed of 5 independent experts. They met in June 2022 and had access to a blinded presentation of the global data from the first 300 patients included by group (self-extubation and repositioning on the ETT). This date has been planned to allow a conclusion to be reached on the safety data before the second period of the study started. Finally, the ISC advised to continue the study without any restriction.

If the study hypothesis is confirmed, the results will provide answers that may change and standardize care practices and make the management of intubated patients in intensive care more rational in this area.

## Study timeline

The current protocol version is V5.0 from November 2022.

Enrolment is ongoing. As of 27/07/2023, 766 patients had been included. Recruitment is expected to be completed by the beginning of August 2023 and the last patient follow-up by the beginning of September. The assessment of the main criterion by the adjudication committee is ongoing and should be completed in the first half of 2024.

## Supporting information

**S1 Checklist. SPIRIT checklist.**
(PDF)

**S1 Protocol. Clinical study protocol.**
(PDF)

**S1 Appendix. IDEFIX study abstract.**
(PDF)

## Acknowledgments

The authors thank Marie-Ange Azaïs for her support at the beginning of the study and Emilie Scalisi for her assistance in preparing and reviewing the manuscript.

## Author Contributions

**Conceptualization:** Vanessa Zinzoni, Lucie Planche, Stéphanie Martin, Jean-Claude Lachérade.

**Funding acquisition:** Vanessa Zinzoni, Stéphanie Martin.

**Investigation:** Vanessa Zinzoni, Sophie Le Potier, Laurence Robin, Cécile Le Parco, Philippe Terrat, Marie-Hélène Leroyer, Romain Atger, Jérôme E. Dauvergne, Lucie Muller, Laetitia Fontaine, Célina Morand, Pascaline Dennemont, Ophélie Paillard, Servane Vastral, Baptiste Dardaine, Sylvie Le Guillou, Natacha Maquigneau.

**Methodology:** Vanessa Zinzoni, Lucie Planche, Stéphanie Martin, Jean-Claude Lachérade.

**Writing – original draft:** Vanessa Zinzoni, Stéphanie Martin.

**Writing – review & editing:** Vanessa Zinzoni, Lucie Planche, Sophie Le Potier, Laurence Robin, Cécile Le Parco, Philippe Terrat, Marie-Hélène Leroyer, Romain Atger, Jérôme E. Dauvergne, Lucie Muller, Laetitia Fontaine, Célina Morand,

Pascaline Dennemont, Ophélie Paillard, Servane Vastral, Baptiste Dardaine,
Sylvie Le Guillou, Natacha Maquigneau, Stéphanie Martin, Jean-Claude Lachérade.

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
