## [Decision Letter · Decision Letter 0]

3 Nov 2023

PONE-D-23-22028Impact of two endotracheal tube fixation on the incidence of peri-oral lesions: elastic adhesive strips versus cord in a protective sheath. Study protocol for a cluster cross-over randomized trial (FIXATUB).PLOS ONE

Dear Dr. zinzoni,

Thank you for submitting your manuscript to PLOS ONE. After careful consideration, we feel that it has merit but does not fully meet PLOS ONE’s publication criteria as it currently stands. Therefore, we invite you to submit a revised version of the manuscript that addresses the points raised during the review process.

We look forward to receiving your revised manuscript.

Kind regards,

Shane Patman, PhD

Academic Editor

PLOS ONE

Journal Requirements:

2. In the ethics statement in the Methods, you have specified that verbal consent was obtained. Please provide additional details regarding how this consent was documented and witnessed, and state whether this was approved by the IRB.

“This study is supported by a grant from the French Ministry of Health (Programme Hospitalier de Recherche Infirmière et Paramédicale 2019, PHRIP-19-0044) obtained by VZ.

Website from French Ministry of Health : https://sante.gouv.fr

The funder has no role in study design, data collection, and analysis, decision to publish, or preparation of the manuscript.”

5. We note that the original protocol that you have uploaded as a Supporting Information file contains an institutional logo. As this logo is likely copyrighted, we ask that you please remove it from this file and upload an updated version upon resubmission.

6. We note that the original protocol file you uploaded contains a confidentiality notice indicating that the protocol may not be shared publicly or be published. Please note, however, that the PLOS Editorial Policy requires that the original protocol be published alongside your manuscript in the event of acceptance. Please note that should your paper be accepted, all content including the protocol will be published under the Creative Commons Attribution (CC BY) 4.0 license, which means that it will be freely available online, and any third party is permitted to access, download, copy, distribute, and use these materials in any way, even commercially, with proper attribution.

Therefore, we ask that you please seek permission from the study sponsor or body imposing the restriction on sharing this document to publish this protocol under CC BY 4.0 if your work is accepted. We kindly ask that you upload a formal statement signed by an institutional representative clarifying whether you will be able to comply with this policy. Additionally, please upload a clean copy of the protocol with the confidentiality notice (and any copyrighted institutional logos or signatures) removed.

Additional Editor Comments (if provided):

Two content expert, experienced peer reviewers have contributed their time and efforts to provide feedback and recommendations to enhance the conduct and reporting of your proposed work, as outlined below.Thank you in anticipation of considering these comments and suggestions in crafting any revision.

Reviewers' comments:

Reviewer's Responses to Questions

**Comments to the Author**

1. Does the manuscript provide a valid rationale for the proposed study, with clearly identified and justified research questions?

Reviewer #1: Yes

Reviewer #2: No

2. Is the protocol technically sound and planned in a manner that will lead to a meaningful outcome and allow testing the stated hypotheses?

Reviewer #1: Yes

Reviewer #2: Yes

3. Is the methodology feasible and described in sufficient detail to allow the work to be replicable?

Reviewer #1: Yes

Reviewer #2: Yes

4. Have the authors described where all data underlying the findings will be made available when the study is complete?

Reviewer #1: Yes

Reviewer #2: No

5. Is the manuscript presented in an intelligible fashion and written in standard English?

Reviewer #1: Yes

Reviewer #2: No

6. Review Comments to the Author

You may also provide optional suggestions and comments to authors that they might find helpful in planning their study.

Reviewer #1: This is a well written and clear paper. The analysis plan is appropriate. As the statistical reviewer I will focus on methods and reporting

Major

1) the power calculations are appropriate and I finally come across a proper power calculation that accounts for the ICC, well done. Having said that, I am worries that the ICC hypothesised is a bit low. more importantly it is customary that more cases are recruited to account for drop outs and missing/poorly recorded data. Can the authors increase the sample a bit to account for these? if the ICC is 0.2, how much do the results change and what is the power then? can the authors provide power with this sample at various plausible ICC levels, e.g. until 0.3 or something? Similarly for different interperiod-class correlations. Also i see that the paper they reference is for continuous outcomes only, unless i'm mistaken. can they provide the code and the software they used for the calculation, in relation to the primary outcome which is binary? can they also double-check their calculations, ignoring the interperiod-class correlation (i.e. assuming a standard cluster randomised design).

Minor

1) can the authors clarify why was the cross-over design considered essential, briefly?

2) alpha level not alpha risk

3) clarify that you will report the observed ICC

4) please refer to inter-class as interperiod-class for clarity

Reviewer #2: Abstract

1. What is ‘FIXATUB’? Please introduce the full-term. Additionally, REA-RAISIN and IDEFIX must be introduced in the main manuscript.

Background

You did not construct paragraphs. That is, it explains the rationale at the sentence level. This gives the reader an unfocused view of the study. Therefore, you must create paragraphs centered around the topic.

1. LINE 75-88 should be the first paragraph.

2. Line 89-101 should be the second paragraph.

3. Line 102-115 may be the third paragraph.

4. Line 116-135, Line 136-146, Line 147-162 may be next paragraphs.

Objectives and study hypotheses

You need to describe your three main objectives and their hypotheses in a clear, descriptive manner. There were no hypotheses for the second and third objectives in this manuscript. Therefore, line 167-170 should be moved to the end of the objectives where the second and third hypotheses are located.

Material and methods

1. What is the difference between inclusion criteria and inclusion? The description of the inclusion criteria seems to be repetitive in inclusion. Or, it could be one of the data collection procedures. If so, you will need data collection procedures such as headings instead of inclusion or follow-up.

2. In the inclusion and follow up section, you need to create several paragraphs, not sentence.

3. Please show two fixation methods using photos.

4. I recommend proposing an eCRF instead of the description in line 334-351 and line 366-376.

5. The descriptions of the data analysis methods are pretty long and distracted. Please make a clear description.

Discussion

1. Does your research focus on patients receiving vasopressors? Line 466 is a very strange expression.

2. I think your discussion should include;

1) Why did you choose cluster cross-over design to compare two fixation methods? What is the strength of your study design?

2) What are the strengths of your study in terms of data collection procedures, outcome measures, and data collection methods?

3. The discussion also needs to be clear in its paragraphs.

7. PLOS authors have the option to publish the peer review history of their article (what does this mean?). If published, this will include your full peer review and any attached files.

Reviewer #1: No

Reviewer #2: No

---

## [Author Response · Author response to Decision Letter 0]

8 Dec 2023

Journal Requirements:

Answers: Following this recommendations, we corrected the Figure citation, suppressed the figures from the manuscript, changed the resolution of the figure and the police…We think we have respected all the requirements.

2. In the ethics statement in the Methods, you have specified that verbal consent was obtained. Please provide additional details regarding how this consent was documented and witnessed, and state whether this was approved by the IRB.

Answer: We do not understand which additional details you want to be added into the manuscript. It is already written that the collection of the oral consent should be documented in the patient’s medical file and that the study protocol and the patient information form was approved by ethics committee. We have change the paragraph to make it clearer. Hoping this will suits you. 

There are no witness. In case the patient is included with the emergency procedure, the relatives and the patient should be informed as soon as possible and should give their approval. If not, patient’s data will not be analysed.

“This study is supported by a grant from the French Ministry of Health (Programme Hospitalier de Recherche Infirmière et Paramédicale 2019, PHRIP-19-0044) obtained by VZ.

Website from French Ministry of Health : https://sante.gouv.fr

The funder has no role in study design, data collection, and analysis, decision to publish, or preparation of the manuscript.”

Please provide an amended statement that declares *all* the funding or sources of support (whether external or internal to your organization) received during this study, as detailed online in our guide for authors at http://journals.plos.org/plosone/s/submit-now. 

Answer: the only funding is the grant from the French Ministry of Health.

Please also include the statement “There was no additional external funding received for this study.” in your updated Funding Statement.

Answer: we have updated the cover letter with the above sentence.

Answer: We added a Supporting Information Files part in the manuscript.

5. We note that the original protocol that you have uploaded as a Supporting Information file contains an institutional logo. As this logo is likely copyrighted, we ask that you please remove it from this file and upload an updated version upon resubmission.

Answer : logo has been removed and protocols have been uploaded.

6. We note that the original protocol file you uploaded contains a confidentiality notice indicating that the protocol may not be shared publicly or be published. Please note, however, that the PLOS Editorial Policy requires that the original protocol be published alongside your manuscript in the event of acceptance. Please note that should your paper be accepted, all content including the protocol will be published under the Creative Commons Attribution (CC BY) 4.0 license, which means that it will be freely available online, and any third party is permitted to access, download, copy, distribute, and use these materials in any way, even commercially, with proper attribution.

Therefore, we ask that you please seek permission from the study sponsor or body imposing the restriction on sharing this document to publish this protocol under CC BY 4.0 if your work is accepted. We kindly ask that you upload a formal statement signed by an institutional representative clarifying whether you will be able to comply with this policy. Additionally, please upload a clean copy of the protocol with the confidentiality notice (and any copyrighted institutional logos or signatures) removed.

Answer : Confidentiality notice and logo have been removed.

A formal statement signed by our director has been attached.

Additional Editor Comments (if provided):

Two content expert, experienced peer reviewers have contributed their time and efforts to provide feedback and recommendations to enhance the conduct and reporting of your proposed work, as outlined below. Thank you in anticipation of considering these comments and suggestions in crafting any revision.

Reviewer #1: 

This is a well written and clear paper. The analysis plan is appropriate. As the statistical reviewer I will focus on methods and reporting

Major

1) the power calculations are appropriate and I finally come across a proper power calculation that accounts for the ICC, well done. Having said that, I am worries that the ICC hypothesised is a bit low. more importantly it is customary that more cases are recruited to account for drop outs and missing/poorly recorded data. Can the authors increase the sample a bit to account for these? if the ICC is 0.2, how much do the results change and what is the power then? can the authors provide power with this sample at various plausible ICC levels, e.g. until 0.3 or something? Similarly for different interperiod-class correlations. Also i see that the paper they reference is for continuous outcomes only, unless i'm mistaken. can they provide the code and the software they used for the calculation, in relation to the primary outcome which is binary? can they also double-check their calculations, ignoring the interperiod-class correlation (i.e. assuming a standard cluster randomised design).

Answer: 

We calculated that the inclusion of 768 patients (24 patients per period per center) would ensure a power of at least 90%. That's why we have chosen not to include an additional percentage of patients. With these same assumptions and for 80% power, 14 patients per center per period are required, i.e. 448 patients in total. In our past experience (DEMETER and HYPERION studies), the rate of patients not meeting inclusion criteria or with a withdrawal of consent should represent 1 to 2% at most. We, therefore, estimate that at least 748 patients will be analyzed (which corresponds to an average inclusion of 23 patients per center per period), which guarantees a power of 89.7%. We can not increase the sample because the recruitment has been terminated after the submission of the manuscript.

As we had no data on the estimation of correlation coefficients, these were defined arbitrarily but in accordance with the recommendations of authors. Thus, we chose a "moderate" intra-class correlation coefficient of 0.1(1). As stated in the paper of Arnup and al (2), it is cautious to set a within-cluster between-period correlation (BPC) lower than the within-cluster within-period correlation (WPC), and suggestions have been made to set the BPC to half the WPC or to 0.8 of the WPC. For the FIXATUB study, we chose to set the BPC to half the WPC, i.e. 0.05.

The number of subjects was calculated using a web-based tool, the Shiny CRT Calculator (3). A cluster size per period of 24 patients for 16 clusters was determined by setting the following parameters: WPC = 0.1, Cluster auto-correlation (CAC) = 0.5, proportion under control = 0.35 and proportion under intervention = 0.20 

As requested by the reviewer, we make simulations for the WPC and then the CAC. 

For a WPC equals to 0.2 and 0.3 (fixing a CAC to 0.5); power drops to 75% and 62% respectively. 

For a WPC equals to 0.1 and a CAC equals to 0.8, the power of the study rises to over 98%; the power rise to over 94% and 90% for a WPC equals to 0.2 and 0.3 respectively. 

Considering a cluster-randomized trial, with a ICC=0.1 and a total number of clusters of 16, the number of subjects required rises to 1664 for 80% power, which would have made the study not realistic.

(1) Hemming K, Taljaard M. Sample size calculations for stepped wedge and cluster randomised trials: a unified approach. J Clin Epidemiol. 2016 Jan;69:137-46. doi: 10.1016/j.jclinepi.2015.08.015. Epub 2015 Sep 5. PMID: 26344808; PMCID: PMC4687983. 

(2) Arnup SJ, McKenzie JE, Hemming K, Pilcher D, Forbes AB. Understanding the cluster randomised crossover design: a graphical illustraton of the components of variation and a sample size tutorial. Trials. 2017 Aug 15;18(1):381. doi: 10.1186/s13063-017-2113-2. PMID: 28810895; PMCID: PMC5557529. 

(3) Hemming K, Kasza J, Hooper R, Forbes A, Taljaard M. A tutorial on sample size calculation for multiple-period cluster randomized parallel, cross-over and stepped-wedge trials using the Shiny CRT Calculator. Int J Epidemiol. 2020 Jun 1;49(3):979-995. doi: 10.1093/ije/dyz237. PMID: 32087011; PMCID: PMC7394950. 

Minor

1) can the authors clarify why was the cross-over design considered essential, briefly?

2) alpha level not alpha risk

3) clarify that you will report the observed ICC

4) please refer to inter-class as interperiod-class for clarity

Answer: We followed your recommendations. The cross-over justification is described in the discussion part.

Reviewer #2: 

Abstract

1. What is ‘FIXATUB’? Please introduce the full-term. Additionally, REA-RAISIN and IDEFIX must be introduced in the main manuscript.

Answer: FIXATUB is an acronym made from FIXation and TUBe. There is no real full-term.

We have deleted it from the manuscript

The term REA-RAISIN has been developed in the manuscript.

IDEFIX is the name of the quoted study. It stands for IDE (state-certified nurse) and FIX for fixation. Unfortunately, we don’t have other way to quote this study as it has not been published. Results were provided during the French Intensive Care congress in 2012. We have added the abstract in supporting file.

Background

You did not construct paragraphs. That is, it explains the rationale at the sentence level. This gives the reader an unfocused view of the study. Therefore, you must create paragraphs centered around the topic.

1. LINE 75-88 should be the first paragraph.

2. Line 89-101 should be the second paragraph.

3. Line 102-115 may be the third paragraph.

4. Line 116-135, Line 136-146, Line 147-162 may be next paragraphs.

Answer: We have followed your recommendations. Thank you for this.

Objectives and study hypotheses

You need to describe your three main objectives and their hypotheses in a clear, descriptive manner. There were no hypotheses for the second and third objectives in this manuscript. Therefore, line 167-170 should be moved to the end of the objectives where the second and third hypotheses are located.

Answer: We have only one main objective (perioral lesions) with one hypothesis. The other are secondary objectives so we did not make any hypothesis about them. Consequently it appears to be more logical to us to let the hypothesis under the main objective.

Material and methods

1. What is the difference between inclusion criteria and inclusion? The description of the inclusion criteria seems to be repetitive in inclusion. Or, it could be one of the data collection procedures. If so, you will need data collection procedures such as headings instead of inclusion or follow-up.

Answer: Inclusion paragraph is a subpart of the part “intervention” in which we describe the study intervention at inclusion. It does not fit with some data collection procedure. We have changed the subtitle to focus on the first use of ETT fixation system. We have deleted some sentences referring to the inclusion criteria and let the sentences referring to study procedure to avoid repetition.

2. In the inclusion and follow up section, you need to create several paragraphs, not sentence.

Answer: We have followed your recommendations.

3. Please show two fixation methods using photos.

Answer: We have added photos.

4. I recommend proposing an eCRF instead of the description in line 334-351 and line 366-376.

Thank you for the remark. However it fits to the study protocol article template provided by Plos One, line “ what outcomes will be measured, when and how”. Besides, when we look at some PLos One protocol articles, all outcome and scales used are described in the text. So we have preferred not to change the manuscript in that way. 

5. The descriptions of the data analysis methods are pretty long and distracted. Please make a clear description.

Answer: To simplify, we have deleted some details that will be described in the Statistical Analysis Plan.

Discussion

1. Does your research focus on patients receiving vasopressors? Line 466 is a very strange expression.

Answer: Indeed, the study focuses on patients receiving vasopressors as it is an inclusion criteria and we wanted to explain why we made this choice. We have rephrased to explain why we focused on this population.

2. I think your discussion should include;

1) Why did you choose cluster cross-over design to compare two fixation methods? What is the strength of your study design

2) What are the strengths of your study in terms of data collection procedures, outcome measures, and data collection methods?

3. The discussion also needs to be clear in its paragraphs.

Answer: We have modified this part to take into account your recommendations.

---

## [Decision Letter · Decision Letter 1]

3 Jan 2024

Impact of two endotracheal tube fixation on the incidence of peri-oral lesions: elastic adhesive strips versus cord in a protective sheath. Study protocol for a cluster cross-over randomized trial.

PONE-D-23-22028R1

Dear Dr. zinzoni,

We’re pleased to inform you that your manuscript has been judged scientifically suitable for publication and will be formally accepted for publication once it meets all outstanding technical requirements.

Kind regards,

Shane Patman, PhD

Academic Editor

PLOS ONE

Additional Editor Comments (optional):

Reviewers' comments:

Reviewer's Responses to Questions

**Comments to the Author**

1. Does the manuscript provide a valid rationale for the proposed study, with clearly identified and justified research questions?

Reviewer #1: Yes

2. Is the protocol technically sound and planned in a manner that will lead to a meaningful outcome and allow testing the stated hypotheses?

Reviewer #1: Yes

3. Is the methodology feasible and described in sufficient detail to allow the work to be replicable?

Reviewer #1: Yes

4. Have the authors described where all data underlying the findings will be made available when the study is complete?

Reviewer #1: Yes

5. Is the manuscript presented in an intelligible fashion and written in standard English?

Reviewer #1: Yes

6. Review Comments to the Author

You may also provide optional suggestions and comments to authors that they might find helpful in planning their study.

Reviewer #1: I am satisfied with the authors' responses and the resulting changes to the paper, I have nothing else to add.

7. PLOS authors have the option to publish the peer review history of their article (what does this mean?). If published, this will include your full peer review and any attached files.

Reviewer #1: No

---

## [Editor Report · Acceptance letter]

30 Jan 2024

PONE-D-23-22028R1 

PLOS ONE

Dear Dr. zinzoni, 

I'm pleased to inform you that your manuscript has been deemed suitable for publication in PLOS ONE. Congratulations! Your manuscript is now being handed over to our production team.

Kind regards, 

on behalf of

Assoc Prof Shane Patman 

Academic Editor

PLOS ONE